# Clinical Utility of Patient-Derived Cell-Based In Vitro Drug Sensitivity Testing for Optimizing Adjuvant Therapy in Dogs with Solid Tumors: A Retrospective Study (2019–2023)

**DOI:** 10.3390/ani15081146

**Published:** 2025-04-16

**Authors:** Young-Rok Kim, Kieun Bae, Ja-Young Lee, Soon-Wuk Jeong, Hun-Young Yoon, Hyun-Jung Han, Jae-Eun Hyun, Aryung Nam, Ji-Hwan Park, Kyong-Ah Yoon, Jung-Hyun Kim

**Affiliations:** 1KU Animal Cancer Center, Konkuk University Veterinary Medical Teaching Hospital, Seoul 05029, Republic of Korea; 2Department of Veterinary Internal Medicine, College of Veterinary Medicine, Konkuk University, Seoul 05029, Republic of Korea; 3Department of Veterinary Biochemistry, College of Veterinary Medicine, Konkuk University, Seoul 05029, Republic of Korea; 4Department of Veterinary Surgery, College of Veterinary Medicine, Konkuk University, Seoul 05029, Republic of Korea; 5Department of Veterinary Emergency and Critical Care, College of Veterinary Medicine, Konkuk University, Seoul 05029, Republic of Korea; 6Bundang Leaders Animal Medical Center, Seongnam 13636, Republic of Korea

**Keywords:** adjuvant therapy, dog, in vitro drug sensitivity testing, personalized therapy, solid tumor

## Abstract

Tumor heterogeneity contributes to inter-individual differences in drug response. In human oncology, patient-derived preclinical models have been proposed to predict individual responses and to select appropriate treatments. Thus, we aimed to evaluate correlations between in vitro drug sensitivity in patient-derived two-dimensional cell cultures and clinical outcomes in dogs with solid tumors receiving postoperative adjuvant therapy. Adjuvant therapy guided by in vitro testing results was associated with a significantly longer time to progression. Our findings indicated that in vitro testing could potentially be used to provide personalized adjuvant therapy and improve clinical outcomes for dogs with solid tumors.

## 1. Introduction

Cancer is the leading cause of death among dogs, with solid tumors comprising most malignancies [1]. The mainstay of treatment for canine solid tumors includes surgery, chemotherapy, radiotherapy, or a combination of these. However, surgery alone is often insufficient even when complete surgical margins are achieved [2]. Therefore, postoperative adjuvant therapy has been proposed to prevent local recurrence and metastasis, with chemotherapy being the most widely used modality in the adjuvant setting. The efficacy of adjuvant chemotherapy has been reported for various solid tumors [3]. However, the narrow therapeutic index and high toxicity associated with chemotherapy limit its use. Moreover, standard protocols may be ineffective owing to intrinsic tumor resistance, and responses to chemotherapy are often unpredictable [4,5].

With an understanding of cancer biology, a paradigm shift has occurred in cancer treatment from conventional chemotherapy to targeted therapy in human and veterinary medicine [4,6]. Compared to cytotoxic drugs that act on rapidly proliferating cells, whether tumor or normal cells, targeted therapy provides selectivity toward tumor cells, minimizing damage to normal cells, by targeting genetic alterations that play crucial roles in tumor growth and progression [7]. In human medicine, targeted therapy can improve patient outcomes in various settings by offering tailored treatments derived from molecular profiling [7,8]. However, limited studies exist on the molecular classification of canine tumors despite their genetic similarities to human malignancies [9]. Furthermore, the relationship between target inhibition and clinical response is not established in canine tumors [10,11]; therefore, targeted therapy has been applied in most veterinary clinics without prior evaluation of molecular profiles. Nevertheless, the clinical efficacy of targeted therapies, mainly tyrosine kinase inhibitors (TKIs), has been reported [12,13,14].

Despite these molecular approaches, several clinical challenges remain unresolved. Inter-individual variations in response to anticancer drugs persist, even when treatments are tailored to molecular profiles, and are largely driven by tumor heterogeneity at the genomic, epigenomic, and microenvironmental levels [15,16]. Therefore, there is an emphasis on the necessity of preclinical models to predict responses to anticancer drugs and provide appropriate treatments to individual patients. Consequently, patient-derived models generated from tumor specimens obtained through surgical resection or biopsies have been developed to evaluate anticancer drug sensitivity [17]. Among various models, two-dimensional (2D) cell culture is the most common platform for high-throughput drug screening owing to advantages of rapidity, technical simplicity, low-cost maintenance, and relatively low cell number [18]. Additionally, cell viability assays, including adenosine triphosphate (ATP)-based and methyl thiazolyl–diphenyl–tetrazolium bromide assays, are essential for assessing anticancer drug cytotoxicity in cell culture models [19].

In human medicine, in vitro drug sensitivity testing has been extensively studied as a precision medicine tool [20]. Numerous studies have highlighted its predictive value by using 2D cell cultures in various human malignancies, such as blood [21], bladder [22], breast [23], gastrointestinal [24,25], head and neck [26], lung [27], and ovarian cancers [28], as well as malignant melanoma [29]. Additionally, postoperative adjuvant chemotherapy, which was found to be sensitive during testing, has been correlated with an extended time to relapse in human patients [30,31,32,33]. However, although there are a few studies on in vitro sensitivity testing in veterinary medicine, these studies were limited to in vitro conditions and rarely evaluated the ability to reflect in vivo conditions [34,35,36,37,38,39,40]. Therefore, we aimed to investigate the clinical utility of in vitro drug sensitivity testing as a predictor of clinical outcomes in dogs with solid tumors. We hypothesized that adjuvant therapy guided by in vitro drug sensitivity test results could improve clinical outcomes compared to empirical treatments.

## 2. Materials and Methods

### 2.1. Ethics Statement

This study was approved by the Institutional Animal Care and Use Committee of the Konkuk University (approval no. KU19189, KU20172, KU21185, KU22180, KU23196). Ethical approval was not required for dogs that did not undergo the in vitro drug sensitivity testing because the research had no impact on therapeutic decisions. Written informed consent for the use of the dogs’ medical records was provided by all owners.

### 2.2. Study Design

This retrospective study was conducted on client-owned dogs with suspected tumors. Medical records from two referral veterinary hospitals (Konkuk University Animal Cancer Center and Bundang Leaders Animal Medical Center) were reviewed for dogs histologically confirmed to have solid tumors, specifically carcinoma or sarcoma, between May 2019 and November 2023. The following data were available for each dog: history, signalment, body weight, blood pressure, hematology, serum chemistry, urinalysis, histopathological findings, immunohistochemical features (if necessary), diagnostic imaging, treatment, and clinical outcomes. All dogs underwent whole body computed tomography before surgery. In most dogs with metastatic disease, suspected lesions were biopsied and confirmed through histopathology by veterinary diagnostic laboratories. Pulmonary metastatic disease in some cases was confirmed only based on imaging features. The TNM classification was evaluated according to the World Health Organization (WHO) criteria for tumors in domestic animals [41]. Time to progression (TTP) was used as the primary endpoint. Adverse events (AEs) were assessed every 1–2 months after the initiation of adjuvant therapy according to the Veterinary Cooperative Oncology Group-Common Terminology Criteria for Adverse Events (VCOG-CTCAE v2.0) [42].

Surgical removal of the primary tumor was performed on all dogs, followed by postoperative adjuvant therapy. The selection of anticancer drugs for adjuvant therapy in each dog was based on either guidance from in vitro drug sensitivity testing (guided group) or histopathological examination alone (empirical group). In the guided group, anticancer drugs, including TKIs and cytotoxic chemotherapeutic agents proved to be effective in individual tumor type, were included in the candidate list for in vitro anticancer drug sensitivity testing. Among the candidate drugs, the anticancer drug that induced more tumor cell death at a lower concentration than others was selected for adjuvant therapy. In contrast, dogs in the empirical group received anticancer drugs based solely on the histopathologic diagnosis and features (Figure 1). To ensure that bias did not influence the results, TTP was measured by a researcher blinded to the treatment group assignment.

Dogs were excluded from the study if they did not receive adjuvant therapy postsurgery, underwent other treatments before surgery, or had surgery with palliative intent. Dogs were also excluded if they received adjuvant therapy lasting <1 month because this duration is not sufficient to evaluate the effect of anticancer drugs [43].

### 2.3. In Vitro Drug Sensitivity Testing

Cells were dissociated from surgically resected tissues through enzymatic digestion with collagenase II (Life Technologies, Carlsbad, CA, USA), hyaluronidase, and Ly27632 (Sigma-Aldrich, St. Louis, MO, USA). After filtering through a 70 µm cell strainer (BD Biosciences, San Diego, CA, USA), the dissociated cells were cultured in advanced Dulbecco’s modified Eagle’s medium/F12 supplemented with 10% fetal bovine serum, 10 mM N-2-hydroxyethylpiperazine-N’-2-ethanesulfonic acid, and GlutaMAX^TM^ (Thermo Fisher Scientific, Waltham, MA, USA), and Zell shield^®^ (Minerva Biolabs, Berlin, Germany). The morphology of these cells was compared with that of the primary tumor tissue cells to ensure that the cells cultured in the 2D system retain the cellular characteristics of the original cancer cells, as evidenced by previous studies [38,44]. After short-term expansion, cells were seeded at 15,000 cells per well in 96-well plates and incubated overnight at 37 °C in a humidified atmosphere of 5% CO_2_. The expansion period, determined by the initial number of cells isolated from the tumor tissue, did not exceed 10 days. After allowing 16–24 h for attachment and settling, the medium was replaced with a medium containing serially dilute candidate drugs once cells reach approximately 80% confluence per well. At least six concentrations ranging from 1 to 100 µM (e.g., 1, 5, 10, 20, 50, 100 µM) were tested for each drug, and vehicle controls of DMSO were included. Each condition was assessed in triplicate, and results were expressed as mean ± standard error. After 24 h of drug exposure, cell viability was measured using the CellTiter-Glo^®^ Luminescent Cell Viability Assay (Promega, Madison, WI, USA). The efficacy of candidate drugs was compared using dose-response curves generated in GraphPad Prism 8 (GraphPad Software, San Diego, CA, USA). Dose-dependent effects were demonstrated by plotting the logarithmic drug concentrations against the relative cell viability (%), normalized to the vehicle control group (Appendix A).

### 2.4. Statistical Analysis

Categorical variables are presented as numbers (percentages); continuous variables are expressed as medians (ranges). Normality of continuous data was assessed using the Shapiro–Wilk test. Categorical variables were compared using Pearson’s chi-square or Fisher’s exact tests. Student’s *t*- or Mann–Whitney U-tests were used to compare continuous variables.

TTP was defined as the time from surgery to disease progression (local recurrence or distant metastasis). Dogs without evidence of recurrence or metastasis until loss to follow-up or at the end of the study were censored for TTP analysis. The Kaplan–Meier product limit method was used to estimate TTP. Survival curves were compared between groups using the log-rank test.

The Cox proportional hazards model was used to evaluate correlations between tumor progression and potential prognostic variables, including age, body weight, sex, neuter status, surgical margin, mitotic count, tumor size, presence or absence of distant metastases at initial diagnosis, time interval from surgery to chemotherapy, cancer treatment type (targeted vs. conventional), and treatment strategy (guided vs. empirical). The mitotic count was converted to a dichotomous variable with a cutoff of 20 per 10 high power fields (HPF) [14,45]; other continuous variables were dichotomized based on the median value as the cut-off point. Variables with a *p*-value < 0.2 in the univariable analysis were included in the multivariate model, and results were presented as hazard ratios (HRs) and 95% CIs. The proportional hazard assumption was verified by visually inspecting log–log plots.

Statistical analyses were conducted using SPSS version 26.0 (IBM Corp., Armonk, NY, USA) and R 4.0.4 software (R Project, Vienna, Austria). Statistical significance was set at *p* < 0.05.

## 3. Results

### 3.1. Patient Characteristics

A total of 126 dogs with suspected tumors were screened; 66 were diagnosed with carcinomas or sarcomas. Among these, 49 dogs’ owners requested in vitro drug sensitivity testing. However, this testing was declined for 10 dogs owing to sudden death during the perioperative period (n = 3) or the financial burden associated with adjuvant therapy (n = 7). Furthermore, cell culture was not possible for five dogs owing to microorganism contamination (n = 3) and an inadequate number of tumor cells (n = 2). These five dogs were moved to the empirical group. Additionally, 18 dogs were excluded for the following reasons: neoadjuvant therapy (n = 2), palliative surgery (n = 2), lack of adjuvant therapy after surgery (n = 5), loss to follow-up after surgery (n = 5), and duration of adjuvant therapy ≤ 1 month (n = 4). As a result, 16 dogs were included in the guided group. Of 22 dogs, including the 5 dogs for whom cell culture was not possible and 17 dogs whose owners did not request testing, 5 dogs were excluded due to absence of adjuvant therapy after surgery (n = 2), loss to follow-up after surgery (n = 2), and a duration of adjuvant therapy of ≤ 1 month (n = 1). Consequently, 17 dogs were included in the empirical group (Figure 2).

Of these 33 dogs, 16 were included in the guided group, and 17 were included in the empirical group. Detailed information on dogs in both groups is provided in Table 1 and Table 2. Hepatocellular carcinoma (HCC) and hemangiosarcoma (HSA) were the most frequent tumor types in both guided and empirical groups. The frequency of these tumors was comparable between the groups (Table 3). The only significant difference was treatment duration, while other baseline characteristics were balanced between the groups (Table 4).

### 3.2. Treatments

In the guided group, three–six anticancer drugs were selected as candidates for in vitro drug sensitivity testing (Appendix A). All dogs in this group were treated with a single agent as determined by the test results, with the majority of dogs receiving TKIs, including toceranib phosphate (n = 13) and imatinib mesylate (n = 1). Similarly, most dogs in the empirical group were treated with a single agent, primarily toceranib, which was the only TKI used (n = 12). Toceranib was administered at a dosage range of 2.4–2.9 mg/kg every other day, which is considered sufficient for target inhibition [46]. Other cytotoxic chemotherapeutic agents were administered at the maximum tolerated dose. The median duration of adjuvant therapy was 150 days (range, 28–1093 days).

### 3.3. Clinical Outcomes and Prognostic Factors

The median follow-up duration was 386 days (range, 55–1121 days). During the follow-up, 8 dogs (50%) in the guided group and 16 dogs (94%) in the empirical group developed local recurrence or distant metastasis. Nine dogs were censored from the TTP analysis because they were lost to follow-up without disease progression or had no evidence of disease progression at the end of the study.

Overall, the median TTP for dogs in the guided and empirical group was 949 days (range: 36–1121 days) and 109 days (range: 37–813 days), respectively. The median TTP for dogs in the guided group was significantly longer than that for dogs in the empirical group (*p* = 0.002) (Figure 3). Subgroup analysis of TTP was performed between the guided and empirical groups (Table 5). Among dogs with carcinoma or sarcoma, only those with sarcoma showed a significantly prolonged median TTP in the guided group compared to that in the empirical group (not reached vs. 144 days; *p* = 0.005). Additionally, a significant difference was noted in median TTP between the two groups in dogs with incomplete surgical margin (949 vs. 109 days; *p* = 0.005), dogs with mitotic count < 20 HPF (949 vs. 105 days; *p* = 0.002), dogs with no evidence of metastatic disease at initial diagnosis (455 vs. 196 days; *p* = 0.002), and dogs receiving TKIs (949 vs. 109 days; *p* = 0.008). For 25 dogs that received toceranib, median TTP was significantly longer in the guided group than in the empirical group (949 vs. 109 days; *p* = 0.018) (Figure 4).

The univariate Cox regression analysis for all dogs with solid tumors revealed that several clinical variables were associated with tumor progression, including the presence of distant metastasis, treatment duration > 150 days, conventional chemotherapy, and empirical treatment. After adjusting for other variables with *p* < 0.2, treatment duration > 150 days (HR, 0.021; 95% CI, 0.002–0.194; *p* = 0.001) and empirical treatment (HR, 4.120; 95% CI, 1.347–12.607; *p* = 0.013) remained significant independent prognostic factors for tumor progression in the multivariate analysis (Table 6).

## 4. Discussion

We compared the clinical outcomes of dogs with solid tumors between two groups based on treatment strategy (guided vs. empirical). Generally, overall survival (OS) is regarded as the most clinically relevant endpoint in clinical trials of cancer treatment. However, the relatively short follow-up time, small population size, and treatments administered after first-line therapy failure may contribute to the lack of statistical power in detecting differences in OS [47]. Therefore, we primarily focused on TTP instead, which is considered reliable for evaluating the response to adjuvant therapy as a surrogate endpoint for OS.

In the present study, all dogs underwent surgical excision of the primary tumor and received anticancer drugs in the adjuvant setting. Although postoperative adjuvant therapy has shown therapeutic benefits in dogs with solid tumors, standard protocols have not been established for several tumor types; furthermore, in some cases, adjuvant therapy may not improve survival outcomes [48,49,50,51]. Therefore, there is a need for in vitro assays to determine optimal adjuvant therapy in both human and veterinary oncology [3,30,31,32,33]. However, in contrast to human oncology, its ability to predict the clinical outcomes of adjuvant therapy in veterinary oncology has not yet been studied. Therefore, we aimed to evaluate correlations between adjuvant therapy guided by in vitro drug sensitivity testing results and clinical outcomes in dogs with solid tumors.

Our study showed that guided adjuvant therapy was associated with an overall improvement in TTP in dogs with solid tumors. TTP in the guided group was significantly longer than that in the empirical group in dogs with incomplete surgical margin and lower mitotic count. However, there was no significant difference found among dogs with complete margins, higher mitotic counts, and conventional chemotherapy. Owing to the small sample size, we cannot rule out the possibility that the statistical power was insufficient to detect significance. Indeed, in previous human studies, adjuvant chemotherapy that was considered sensitive in in vitro assays improved clinical outcomes in patients who underwent curative resection [31,32] and in patients with advanced cancer [32].

For dogs with sarcomas, those in the guided group exhibited a longer TTP. Among the subtypes of sarcomas, HSA was more frequent in the empirical group (23.5%) than in the guided group (12.5%). Owing to its biologically aggressive nature, adjuvant therapy is necessary in dogs with HSA, and the time interval between surgery and adjuvant therapy is associated with prognosis. One study found that the administration of adjuvant therapy within 21 days improved OS [52]. In our study, it took approximately 3 weeks to confirm the results of histopathologic examination and in vitro drug sensitivity testing. Consequently, more dogs with HSA may have been included in the empirical group, potentially contributing to the worse outcome. Nevertheless, no statistically significant difference was observed in the frequency of HSA between groups. Further large-scale studies are necessary to confirm this result.

In both the guided and empirical groups, HCC was the most frequent subtype among carcinomas. For canine HCC, surgical removal is the primary treatment option, and surgery alone can provide a good prognosis. However, adjuvant therapy could be considered if the tumor is incompletely resected [53]. Nevertheless, the efficacy of adjuvant therapy for canine HCC remains unclear, and the massive form of HCC has a low recurrence rate (0–13%), even after incomplete resection [54]. In our study, all dogs with HCC had incomplete surgical margins, with more than half (71.4%) occurring in the massive form. Consequently, adjuvant therapy, with or without guidance of in vitro assays, may not have been associated with tumor progression. This could explain why no significant difference was noted in TTP between the two groups among dogs with carcinomas.

Guided adjuvant therapy also significantly increased TTP in dogs without distant metastasis at the initial diagnosis. However, no improvement in TTP was observed in dogs with distant metastases when treated with guided treatments. Metastasis is a multifaceted and complex process driven by the acquisition of genetic and epigenetic alterations within tumor cell and its microenvironment [55]. Therefore, metastatic tumors are genetically similar but are not identical to primary tumors [15,56]. Our attempt to apply treatment based on results obtained from the primary tumors may have led to the selection of inappropriate anticancer drugs for metastatic disease, resulting in undesirable outcomes.

Interestingly, dogs receiving TKIs guided by the in vitro test exhibited a significantly longer TTP than those treated empirically. In veterinary oncology, TKIs such as toceranib, imatinib, and sorafenib have been commonly used for treating dogs with solid tumors and are more effective than conventional chemotherapy [11,57,58]. However, the treatment response has not consistently correlated with the mutational status of tumor tissues [59,60]. Similarly, heterogeneous responses to TKIs have been observed in human patients with the identical driver mutation [16]. Our data could provide evidence that in vitro drug sensitivity testing could optimize TKI efficacy.

Our study had some limitations, mainly related to its retrospective nature. First, the small sample size may predispose the subgroup analysis to both type 1 and 2 statistical errors. Second, the short follow-up duration may have been insufficient to identify tumor recurrence, despite using TTP as the primary endpoint. Third, the selection of candidates for in vitro tests was dependent on clinicians; therefore, inconsistent candidate lists were adopted even among dogs with the same histopathologic diagnosis. In the empirical group, owners favored TKIs over conventional chemotherapy because of their oral bioavailability and low toxicity related to their specificity for tumor cells. Previous studies have reported a lower incidence of AEs in dogs treated with TKIs compared to those treated with conventional chemotherapy [11,57]. Similarly, in our study, 19 of 26 dogs (73.1%) receiving TKIs experienced AEs, with only 1 (5.3%) having grade 3 or 4 toxicity. In contrast, all seven dogs (100%) receiving conventional chemotherapy experienced AEs, with grade 3 or 4 toxicities reported in three (42.9%) dogs (Appendix A). Lastly, the status of regional lymph nodes was not evaluated using fine-needle aspiration or excisional biopsy. Since lymph node metastasis is a potential prognostic factor for various types of solid tumors in dogs, the lack of this information may impact the interpretation of our results [61]. Nevertheless, our study provides preliminary evidence for the feasibility of patient-derived cell-based in vitro drug sensitivity testing in veterinary oncology.

## 5. Conclusions

To our knowledge, this is the first study to report the clinical utility of in vitro drug sensitivity testing for optimizing adjuvant therapy in dogs with solid tumors. Our study found that drug sensitivity in a patient-derived 2D cell culture is positively correlated with clinical outcomes. Therefore, anticancer drugs guided by in vitro drug sensitivity testing results could be optimal adjuvant therapy for dogs with solid tumors. This therapeutic approach also provides alternative options for tumors for which standard protocols are unavailable. Although large-scale randomized controlled trials and further validation for specific tumor types are needed, our findings serve as a valuable foundation for future research.

## Figures and Tables

**Figure 1 animals-15-01146-f001:**
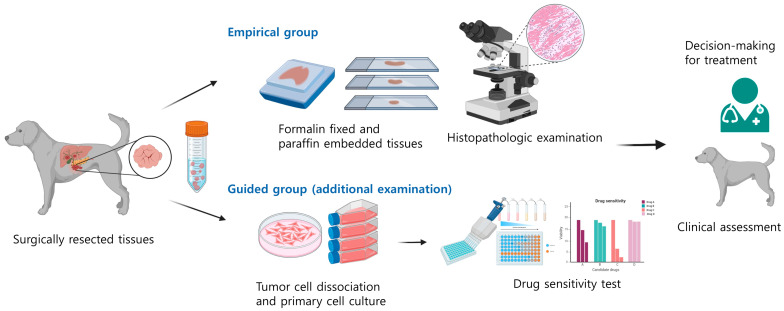
Overview of the selection of anticancer drugs for dogs. In the guided group, tumor tissues were surgically resected and used to establish a primary cell culture. The list of candidate drugs comprised TKIs and cytotoxic chemotherapeutic agents, which have been proven to be effective in specific tumor types. The anticancer drug that induced tumor cell death more effectively at a lower concentration was chosen. In contrast, anticancer drug was determined solely based on histopathologic diagnosis.

**Figure 2 animals-15-01146-f002:**
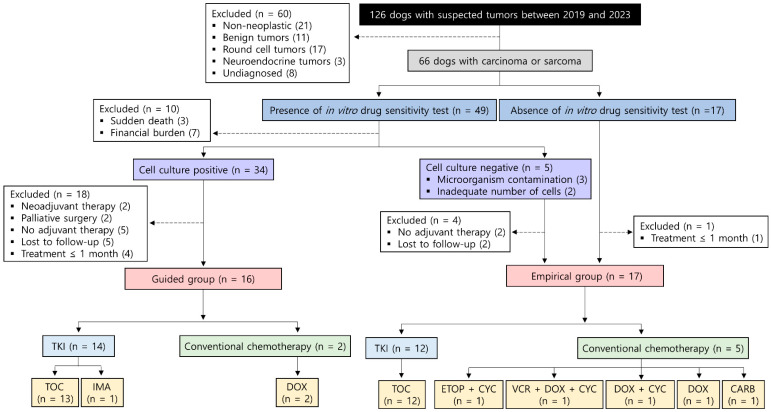
Flow chart of the study population. Abbreviations: CARB, carboplatin; CYC, cyclophosphamide; DOX, doxorubicin; ETOP, etoposide; IMA, imatinib; TKI, tyrosine kinase inhibitor; TOC, toceranib; VCR, vincristine.

**Figure 3 animals-15-01146-f003:**
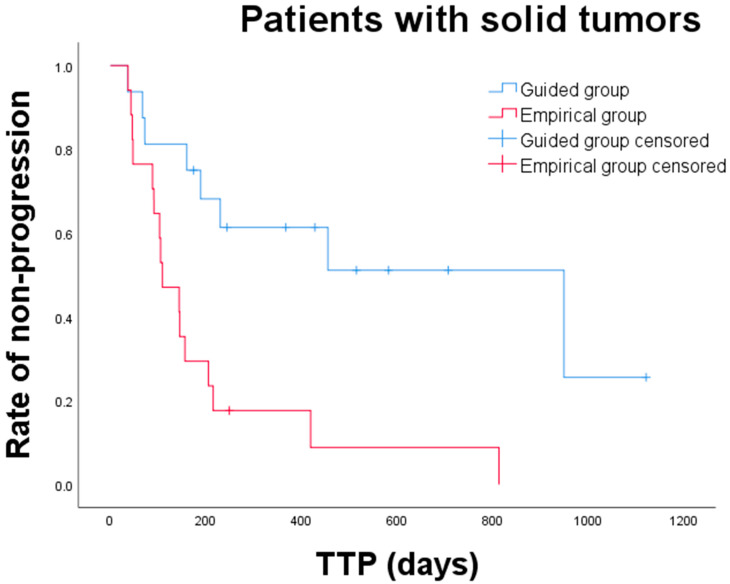
Kaplan–Meier curves for time to progression (TTP) of dogs with solid tumors between guided and empirical groups. Dogs in the guided group (blue line, n = 16) had a significantly longer TTP than those in the empirical group (red line; n = 17) (*p* = 0.002).

**Figure 4 animals-15-01146-f004:**
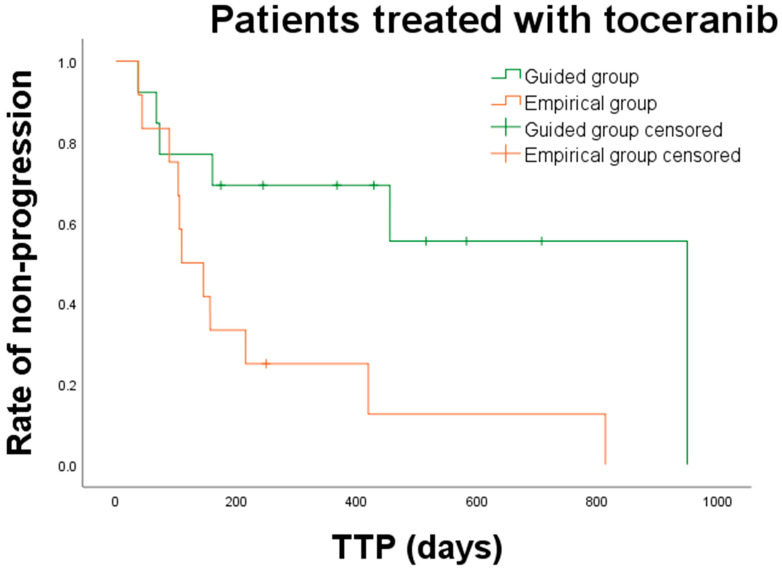
Kaplan–Meier curves for time to progression (TTP) in dogs treated with toceranib in the guided and empirical group. A significant difference in TTP was observed between guided (green line, n = 13) and empirical (orange line, n = 12) groups (*p* = 0.018).

**Table 1 animals-15-01146-t001:** Information of each dog in the guided group.

Case	Breed	Age (Years)	Sex	Tumor Location	Diagnosis	TNM	Anticancer Drugs	Treatment Duration (Days)	TTP (Days) ^a^
1	Pomeranian	10	SF	Liver	HCC-CC	T1N0M0	Toceranib	734	949
2	Mixed	11	IF	Liver	HCC-CC	T3N0M0	Toceranib	685	455
3	Cocker spaniel	8	CM	Liver	HCC	T1N0M0	Toceranib	93	174 ^b^
4	Mixed	9	CM	Liver	HCC	T1NXM0	Toceranib	172	582 ^c^
5	Bichon frise	10	SF	Liver	HCC	T1N0M0	Toceranib	376	428 ^c^
6	Scottish terrier	8	CM	Liver	HCC	T1N0M0	Toceranib	323	367 ^b^
7	Maltese	5	CM	Kidney	RCC	T3NXM1	Toceranib	41	36
8	Chihuahua	6	SF	Kidney	RCC	T3NXM1	Toceranib	142	72
9	Pomeranian	5	SF	Oral cavity	SCC	T1N1M0	Toceranib	143	160
10	French bulldog	10	SF	Lung	PC	T1NXM0	Doxorubicin	183	189
11	Great Pyrenees	12	CM	Muscle	HSA	T3N0M0	Toceranib	79	67
12	Yorkshire terrier	13	SF	Spleen	HSA	T1N0M0	Doxorubicin	264	230
13	Shih tzu	15	SF	Skin	STS	T1N0M0	Toceranib	651	707 ^b^
14	Maltese	11	SF	Intestine	GIST	T1N0M0	Imatinib	1093	1121 ^c^
15	Coton de Tulear	7	CM	Lung	HS	T1N0M0	Toceranib	197	244 ^c^
16	Golden retriever	6	CM	Bone	OSA	T2N0M0	Toceranib	471	515 ^b^

Note: The TNM classification was evaluated according to the World Health Organization (WHO) criteria for tumors in domestic animals. Abbreviations: CM, castrated male; IF, intact female; SF, spayed female; HCC, hepatocellular carcinoma; HCC-CC, combined hepatocellular cholangiocarcinoma; HS, histiocytic sarcoma; HSA, hemangiosarcoma; GIST, gastrointestinal stromal tumor; OSA, osteosarcoma; PC, pulmonary carcinoma; RCC, renal cell carcinoma; SCC, squamous cell carcinoma; STS, soft tissue sarcoma. ^a^ TTP was defined as the time from surgery to either disease progression; ^b^ patients without disease progression at the end of the study; ^c^ patients lost to follow-up without disease progression.

**Table 2 animals-15-01146-t002:** Information of each dog in the empirical group.

Case	Breed	Age (Years)	Sex	Tumor Location	Diagnosis	TNM	Anticancer Drugs	Treatment Duration (Days)	TTP (Days) ^a^
1	Shih tzu	12	CM	Liver	HCC	T2N0M0	Toceranib	78	103
2	Poodle	7	IF	Liver	HCC	T1N0M0	Toceranib	593	813
3	Poodle	11	CM	Liver	HCC	T1N0M0	Toceranib	227	249 ^b^
4	Japanese spitz	9	CM	Lung	PC	T1NXM1	Toceranib	130	109
5	Pekingese	14	SF	Lung	PC	T1N0M0	Toceranib	29	43
6	Maltese	6	SF	Anal sac	AGASACA	T3N0M0	Toceranib	264	419
7	Silky terrier	10	CM	Intestine	SBA	T2NXM1	Toceranib	145	145
8	Maltese	10	SF	Pancreas	PAC	T1N1M0	Toceranib	133	105
9	Poodle	8	CM	Tonsil	SCC	T2N1M0	Toceranib	123	88
10	Italian greyhound	3	IF	Ovary	OC	T3N0M0	Carboplatin	176	205
11	Bichon frise	11	CM	Nasal cavity	NC	T2N0M0	Toceranib	235	156
12	French bulldog	10	CM	Spleen	HSA	T2N0M0	ETOP + CYC	150	144
13	Mixed	11	CM	Spleen	HSA	T3NXM1	Doxorubicin	29	46
14	Maltese	9	SF	Spleen	HSA	T2NXM0	VCR + DOX + CYC	28	91
15	Maltese	10	CM	Spleen	HSA	T2N0M1	DOX + CYC	70	47
16	Welsh corgi	10	CM	Oral cavity	STS	T1NXM0	Toceranib	40	37
17	Beagle	8	SF	Intestine	GIST	T2NXM1	Toceranib	196	215

Note: The TNM classification was evaluated according to the World Health Organization (WHO) criteria for tumors in domestic animals. Abbreviations: CM, castrated male; IF, intact female; SF, spayed female; AGASACA, apocrine gland anal sac adenocarcinoma; GIST, gastrointestinal stromal tumor; HCC, hepatocellular carcinoma; HSA, hemangiosarcoma; NC, nasal carcinoma; OC, ovarian carcinoma; PC, pulmonary carcinoma; PAC, pancreatic adenocarcinoma; SBA, small bowel adenocarcinoma; SCC, squamous cell carcinoma; STS, soft tissue sarcoma; CYC, cyclophosphamide; DOX, doxorubicin; ETOP, etoposide; VCR, vincristine. ^a^ TTP was defined as the time from surgery to either disease progression; ^b^ Patients without disease progression at the end of the study.

**Table 3 animals-15-01146-t003:** Distribution of tumor subtypes between the guided and empirical groups.

Tumor Types (n)	Guided Group (n = 16)	Empirical Group (n = 17)
Carcinoma (n = 21)		
Hepatocellular carcinoma	4 (25.0%)	3 (17.6%)
Pulmonary carcinoma	1 (6.25%)	2 (11.8%)
Squamous cell carcinoma	1 (6.25%)	1 (5.9%)
Combined hepatocellular cholangiocarcinoma	2 (12.5%)	-
Renal cell carcinoma	2 (12.5%)	-
Small bowel adenocarcinoma	-	1 (5.9%)
Apocrine gland anal sac adenocarcinoma	-	1 (5.9%)
Pancreatic adenocarcinoma	-	1 (5.9%)
Ovarian carcinoma	-	1 (5.9%)
Nasal carcinoma	-	1 (5.9%)
Total	10	11
Sarcoma (n = 13)		
Hemangiosarcoma	2 (12.5%)	4 (23.5%)
Gastrointestinal stromal tumor	1 (6.25%)	1 (5.9%)
Soft tissue sarcoma	1 (6.25%)	1 (5.9%)
Osteosarcoma	1 (6.25%)	-
Histiocytic sarcoma	1 (6.25%)	-
Total	6	6

**Table 4 animals-15-01146-t004:** Baseline characteristics between the guided and empirical groups.

Variables	Guided Group (n = 16)	Empirical Group (n = 17)	*p*
Median age (years) (range)	9.5 (5–15)	10 (3–14)	0.811
Median body weight (kg) (range)	7.1 (2.0–54.0)	5.8 (3.3–16.5)	0.260
Sex (n) Male Female	7 (43.8%) 9 (56.3%)	10 (58.8%) 7 (41.2%)	0.387
Neuter status (n) Neutered Intact	15 (93.8%) 1 (6.3%)	15 (88.2%) 2 (11.8%)	1.000
Tumor types (n) Carcinoma Sarcoma	10 (62.5%) 6 (37.5%)	11 (64.7%) 6 (35.3%)	0.895
Surgical margin (n) Complete Incomplete	5 (31.3%) 11 (68.8%)	6 (35.3%) 11 (64.7%)	0.805
Mitotic count (per 10 HPF) (n) <20 ≥20	11 (68.8%) 5 (31.3%)	14 (82.4%) 3 (17.6%)	0.438
Median tumor size (mm) ^a^ (range)	50.4 (11.2–125.0)	55.0 (26.0–139.0)	0.443
Distant metastasis (n) Present Absent	2 (12.5%) 14 (87.5%)	5 (29.4%) 12 (70.6%)	0.398
Time interval from surgery to chemotherapy (n) ≤28 days >28 days	9 (56.3%) 7 (43.8%)	13 (76.5%) 4 (23.5%)	0.218
Median treatment duration (days) (range)	231 (41–1093)	133 (28–593)	0.037 *
Types of cancer treatment (n) Targeted therapy (i.e., TKIs) Conventional chemotherapy	14 (87.5%) 2 (12.5%)	12 (70.6%) 5 (29.4%)	0.398

Abbreviation: HPF, high power field; LNs, lymph nodes; TKIs, tyrosine kinase inhibitors. ^a^ Tumor size was calculated as the longest diameter. * Statistically significant (*p* < 0.05).

**Table 5 animals-15-01146-t005:** Subgroup analysis of TTP between the guided and empirical groups.

Variables	Median TTP (Range) (Days)				
	Guided Group	N	Empirical Group	N	*p*
Total (n = 33)	949 (36–949)	16	109 (37–813)	17	0.002 *
Tumor types Carcinoma (n = 21) Sarcoma (n = 12)	455 (36–949) NR (67–230)	10 6	145 (43–813) 144 (37–215)	11 6	0.084 0.005 *
Surgical margin Complete Incomplete	230 (36–230) 949 (67–949)	5 11	91 (46–419) 109 (37–813)	6 11	0.232 0.005 *
Mitotic count (per 10 HPF) <20 ≥20	949 (67–949) 160 (36–160)	11 5	105 (37–813) 109 (91–144)	14 3	0.002 * 0.331
Distant metastasis Present (n = 7) Absent (n = 26)	67 (67–72) 455 (36–949)	2 14	109 (46–215) 196 (37–813)	5 12	0.429 0.002 *
Types of cancer treatment TKIs (n = 26) Conventional chemotherapy (n = 7)	949 (36–949) 189 (189–230)	14 2	109 (37–813) 91 (46–205)	12 5	0.008 * 0.153

Abbreviation: CI, confidence interval; NR, not reached; HPF, high power field; TKIs, tyrosine kinase inhibitors; TTP, time to tumor progression. * Statistically significant (*p* < 0.05).

**Table 6 animals-15-01146-t006:** Univariable and multivariable Cox regression analysis of variables potentially associated with tumor progression in dogs with solid tumors.

Variables	Tumor Progression (Total) (n = 33)			
	Univariable Analysis		Multivariable Analysis	
	Hazard Ratio (95% CI)	*p*	Hazard Ratio (95% CI)	*p*
Age ≥ 10 years	1.191 (0.520–2.724)	0.679	-	-
Weight ≥ 6 kg	0.941 (0.406–2.180)	0.887	-	-
Female sex	0.694 (0.298–1.620)	0.399	-	-
Intact status	0.839 (0.246–2.869)	0.780	-	-
Incomplete margin	0.588 (0.250–1.383)	0.224	-	-
Mitotic count ≥ 20/10 HPF	1.569 (0.609–4.042)	0.351	-	-
Tumor size ≥ 53 mm	1.664 (0.726–3.815)	0.229	-	-
Distant metastasis	4.013 (1.531–10.517)	0.005 *	1.565 (0.517–4.739)	0.428
Time interval from surgery to chemotherapy > 28 days	0.435 (0.160–1.180)	0.102	0.779 (0.258–2.350)	0.657
Treatment duration > 150 days	0.022 (0.003–0.177)	<0.001	0.021 (0.002–0.194)	0.001 *
Conventional chemotherapy	2.571 (1.015–6.510)	0.046 *	1.458 (0.509–4.178)	0.482
Empirical treatment	3.861 (1.549–9.628)	0.004 *	4.120 (1.347–12.607)	0.013 *

Note: Variables with statistically significant association (*p* < 0.2) on univariable analysis were included in the multivariable model. Abbreviations: HPF, high power field; CI, confidence interval. * Statistically significant (*p* < 0.05).

## Data Availability

The data that support the findings of this study are available from the corresponding author upon reasonable request.

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
