# Peer review of "Clinical Utility of Patient-Derived Cell-Based In Vitro Drug Sensitivity Testing for Optimizing Adjuvant Therapy in Dogs with Solid Tumors: A Retrospective Study (2019–2023)"

_animals, 2025, doi:10.3390/ani15081146_

Round 1
Reviewer 1 Report (New Reviewer)
Comments and Suggestions for Authors
This study aims to evaluate the potential benefits of in vitro drug-sensitivity testing using a patient-derived cell-culture model for dogs with solid tumors. It compares this approach to conventional empirical treatment methods, focusing on whether it can improve clinical outcomes, particularly in terms of time to tumor progression (TTP).
By evaluating the therapeutic potential of in vitro drug-sensitivity testing in veterinary oncology, particularly for dogs with solid tumors, this study contributes to advancing the field. While drug-sensitivity testing has been explored extensively for cancer treatment in humans, its application in veterinary medicine, especially for dogs, remains limited. This research aims to bridge that gap, offering new insights into personalized treatment approaches for canine cancer patients.
Selection bias may be introduced due to the observational study design, as dogs are assigned to either the guided or empirical treatment groups based on existing medical practices. A randomized controlled trial (RCT) would strengthen the causal inference between drug-sensitivity testing and improved outcomes by minimizing bias and ensuring more reliable comparisons between treatment approaches.
The study does not mention blinding, which could impact outcomes. If researchers or clinicians are aware of the treatment type (empirical or guided), it may introduce bias in treatment decisions and data interpretation, potentially skewing results.
The follow-up duration should be sufficient to capture any late-onset therapeutic side effects or tumor recurrence. Although the study reports time to tumor progression (TTP), a longer follow-up period is essential to evaluate long-term survival, recurrence, and the treatment's safety and effectiveness over time. This would provide a more thorough and comprehensive assessment of the treatment's overall impact.
The empirical group’s treatment is based on histopathological examination, but the study does not specify whether a standard protocol was followed for evaluating tumor characteristics, such as mitotic count, surgical margin status, and histological grade. Clarifying this would ensure consistency and reliability in the treatment decision-making process.
Standardization of the in vitro drug-sensitivity testing process, including the drug concentrations, incubation times, and assessment of drug response, are essential to ensure consistency across cases.
The study does not explore why certain medications are more effective than others in specific cases, which is a critical factor for understanding the mechanisms behind treatment responses. Investigating these variations would offer valuable insights into how treatments work, helping to identify the factors that contribute to treatment success or failure in individual patients. This understanding could ultimately guide more personalized and effective therapeutic strategies.
The methodology has a solid foundation, with a clear experimental design and relevant outcome measures. However, there are areas for improvement, including:
- Larger sample sizes for each group.
- Clearer description of the treatment selection process and possible confounding factors.
- More detailed statistical analysis and consideration of additional outcome measures.
- Providing more transparency regarding diagnostic test accuracy.
The conclusions are consistent with the evidence presented in the study and directly address the main research question. The observed positive correlation between in vitro drug-sensitivity testing and improved clinical outcomes supports the argument that this approach can enhance adjuvant therapy for dogs with solid tumors. However, the authors rightly emphasize the need for further research, including larger-scale, randomized controlled trials, to fully validate these findings and explore the broader applicability of this approach across different tumor types
Other comments as per the attached file

Author Response
Comments : The study does not mention blinding, which could impact outcomes. If researchers or clinicians are aware of the treatment type (empirical or guided), it may introduce bias in treatment decisions and data interpretation, potentially skewing results.
|
Response: We thank the reviewer for comments. Due to the retrospective nature of our study, group allocation was determined in the past. Therefore, blinding was not feasible, as the researcher cannot control the assignment of participants to treatment groups at the time of treatment initiation. However, in retrospective studies, there are methods to implement blinding in order to reduce bias or ensure that the interpretation of results is not influenced by the researcher. The outcome assessor can be influenced by knowing which treatment was given, so it is essential to keep them blinded. In our study, the researcher measured time to tumor progression (TTP) without knowing which treatment group the dogs belonged to, ensuring that personal preferences or biases did not influence the results. We have added the sentence as follows. Line 130-131: “To ensure that bias did not influence the results, TTP was measured by a researcher blinded to the treatment group assignment.”
|
Comments : The follow-up duration should be sufficient to capture any late-onset therapeutic side effects or tumor recurrence. Although the study reports time to tumor progression (TTP), a longer follow-up period is essential to evaluate long-term survival, recurrence, and the treatment's safety and effectiveness over time. This would provide a more thorough and comprehensive assessment of the treatment's overall impact.
|
Response: We thank the reviewer for comments. As the reviewer rightly pointed out, the follow-up time in our study was relatively short (median, 386 days) and may have insufficient to identify recurrence or long-term survival in this population, despite using TTP as the primary endpoint. Nevertheless, this study provides preliminary evidence for the feasibility of in vitro drug sensitivity test in veterinary oncology. Therefore, we have added this explanation in limitation to justify the short follow-up period as a background for a more extensive study. Line 379-380: “Second, the short follow-up duration may have been insufficient to identify tumor recurrence, despite using TTP as the primary endpoint.” Line 392-394: “Nevertheless, our study provides preliminary evidence for the feasibility of patient-derived cell-based in vitro drug sensitivity testing in veterinary oncology.”
Comments : The empirical group’s treatment is based on histopathological examination, but the study does not specify whether a standard protocol was followed for evaluating tumor characteristics, such as mitotic count, surgical margin status, and histological grade. Clarifying this would ensure consistency and reliability in the treatment decision-making process.
|
Response: We thank the reviewer for comments. The treatment in the empirical group is indeed based on histopathological examination, including diagnosis and characteristics. To improve clarity, we have revised the sentence as follows. Line 129-130: “In contrast, dogs in the empirical group received anticancer drugs based solely on the histopathologic diagnosis and features (Figure 1).”
|
Comments : Standardization of the in vitro drug-sensitivity testing process, including the drug concentrations, incubation times, and assessment of drug response, are essential to ensure consistency across cases.
|
Response: We appreciate the reviewer’s insightful comment. We have revised the methods (section 2.3) to clarify the in vitro drug-sensitivity testing process. We evaluated the dose-dependent effects of candidate drugs by testing at least six concentrations under the condition of 80% confluence of cells. To compare the efficacy of candidate drugs, we generated dose-response curves using GraphPad Prism software, in which the logarithmic drug concentrations were plotted against the relative cell viability (%) normalized to vehicle control-treated cells as shown in supplementary materials (Table S1). These modifications have been added to the Materials and Methods section as follows. Line 129-130: “After allowing 16–24 hours for attachment and settling, the medium was replaced with a medium containing serially dilute candidate drugs once cells reach approximately 80% confluence per well. At least six concentrations ranging from 1 to 100 µM (e.g., 1, 5, 10, 20, 50, 100 µM) were tested for each drug, and vehicle controls of DMSO were included. Each condition was assessed in triplicate, and results were expressed as mean ± standard error. After 24 hours of drug exposure, cell viability was measured using the CellTiter-Glo® Luminescent Cell Viability Assay (Promega, Madison, WI, USA). The efficacy of can-didate drugs was compared using dose-response curves generated in GraphPad Prism (GraphPad Software, San Diego, CA, USA). Dose-dependent effects were demonstrated by plotting the logarithmic drug concentrations against the relative cell viability (%), normalized to the vehicle control group (Table S1).”
|
Comments : The study does not explore why certain medications are more effective than others in specific cases, which is a critical factor for understanding the mechanisms behind treatment responses. Investigating these variations would offer valuable insights into how treatments work, helping to identify the factors that contribute to treatment success or failure in individual patients. This understanding could ultimately guide more personalized and effective therapeutic strategies.
|
Response: We thank the reviewer for comments. Cytotoxic chemotherapeutic agents and tyrosine kinase inhibitors (TKIs), which were used in our study, exert their effects through distinct mechanisms and are applied to specific tumor types accordingly. However, as described in the ‘Introduction’ section, variations in treatment response are frequently observed due to tumor heterogeneity, making it difficult to attribute efficacy solely to known mechanisms. Given this complexity, our study aims to indirectly assess which anticancer agents may be effective in vitro, rather than identifying a specific mechanistic basis for their activity. Nonetheless, this approach could serve as a foundation for understanding the inherent variability in tumor biology.
|
Comments : The methodology has a solid foundation, with a clear experimental design and relevant outcome measures. However, there are areas for improvement, including: - Larger sample sizes for each group. - Clearer description of the treatment selection process and possible confounding factors. - More detailed statistical analysis and consideration of additional outcome measures. - Providing more transparency regarding diagnostic test accuracy.
|
Response: We thank the reviewer for comments. Most of the suggestions made by the reviewer correspond to the limitation of our study. 1) Although large-scale randomized controlled trials are required, our study provide preliminary evidence for the feasibility of in vitro drug sensitivity in veterinary oncology and serve as foundation for further research. 2) Furthermore, due to the retrospective nature of our study, group allocation was determined in the past. As a result, blinding was not feasible, as the researcher cannot control the assignment of participants to treatment groups at the time of treatment initiation. Instead, TTP was measured by a researcher blinded to the treatment group assignment to ensure that bias, possible confounding factors, did not influence the results. 3) Additionally, our study only compares TTP between the groups due to the short follow-up time. However, follow-up studies will allow for consideration of additional outcome measures (e.g., overall survival). 4) In our study, only diagnostic test was histopathological examination, which was confirmed by veterinary diagnostic laboratories. If we have not fully understood the reviewer’s comment, we would appreciate further clarification, and we will address it accordingly.
|
Comments : The conclusions are consistent with the evidence presented in the study and directly address the main research question. The observed positive correlation between in vitro drug-sensitivity testing and improved clinical outcomes supports the argument that this approach can enhance adjuvant therapy for dogs with solid tumors. However, the authors rightly emphasize the need for further research, including larger-scale, randomized controlled trials, to fully validate these findings and explore the broader applicability of this approach across different tumor types.
|
Response: We thank the reviewer for comments. We have already described the need for large-scale randomized controlled trials to strength our results in the ‘Conclusions’section. However, as the reviewer suggested, we have revised the following sentence to acknowledge the need for further research, while emphasizing the significance and value of our study. Line 403-404: “Although large-scale randomized controlled trials and further validation for specific tumor types are needed, our findings serve as a valuable foundation for future research.”
|
Line 65 : delete ‘well’
|
Response: We thank the reviewer for comments. We revised the sentence as follows. Line 64-65: “Furthermore, the relationship between target inhibition and clinical response is not established in canine tumors”
|
Line 201-204 : Not clear and confusing.
|
Response: We thank the reviewer for comments. To improve clarity, we revised the sentence as follows. Line 207-211: “Of 22 dogs, including the five dogs for whom cell culture was not possible and 17 dogs whose owners did not request testing, five dogs were excluded due to absence of adjuvant therapy after surgery (n = 2), loss to follow-up after surgery (n = 2), and duration of adjuvant therapy of ≤ 1 month (n = 1). Consequently, 17 dogs were included in the empirical group (Figure 2)” |
Reviewer 2 Report (Previous Reviewer 1)
Comments and Suggestions for Authors
I thank the authors for revising the manuscript and responding point-by-point to my comments on the previous version. I have confirmed that the manuscript has been sufficiently improved.
Author Response
Comments: I thank the authors for revising the manuscript and responding point-by-point to my comments on the previous version. I have confirmed that the manuscript has been sufficiently improved.
|
Response: We thank the reviewer for their comments. We appreciate the constructive feedback provided in the first round, which has helped improve the manuscript. |
This manuscript is a resubmission of an earlier submission. The following is a list of the peer review reports and author responses from that submission.
Round 1
Reviewer 1 Report
Comments and Suggestions for Authors
As cancer is the leading cause of death among dogs, the authors aimed to evaluate the clinical utility of in vitro drug-sensitivity testing using a patient-derived cell-culture model to select appropriate adjuvant therapies for dogs with solid tumors.Although the aim of the study is worthwhile, there are serious concerns to be published as a research article.
1. Although major findings of this paper (Figures 3 an 4) are on the basis of the comparison of mixing different tumor types (HCC, HSA, etc.) between guided and empirical groups, the progression of the disease is greatly affected by the types and authors should compared within specific tumors (eg. Boston et. al. 2014. Efficacy of systemic adjuvant therapies administered to dogs after excision of oral malignant melanomas: 151 cases (2001-2012). J Am Vet Med Assoc 245: 401-407). 2. The possibility that there was a difference in the severity of the disease condition between the two groups compared (guided and empirical) cannot be ruled out. In fact, the authors mention the impact of distant metastasis in this study from line 327, and as shown in tables 1 and 2, cases of distant metastasis are more in the guided (2) than in the empirical (5). No statistical significance was observed for the median tumor sizes between guided and empirical groups, but the percentage of large tumors (T2 and T3) is higher in the empirical group than in the guided group according to the TNM classification system. 3. Authors describe in lines 179-181, "In the guided group, cell culture was not possible for 5 dogs owing to microorganism contamination (n = 3) and an inadequate number of tumor cells (n = 2). Consequently, these dogs were moved to the empirical group". The possibility cannot be ruled out that the change of groups in these cases may have influenced the study conclusions. These cases should be removed from the study. Acquisition of enough cell number may be related to the tumor characteristics. Contamination of cultured cells cannot be ruled out as a complication of tumor and infection. 4. The method of the patient-derived cell-based in vitro drug-sensitivity testing should be described more clearly for the clinical use. For example, criteria for the selection of the drug is unclear in spite of the description in lines 120-121, the anticancer drug that induced more tumor cell death at a lower concentration than others was selected for adjuvant therapy. 5. The method of the expansion of the culture should be described in detail. Authors describe in line 143 “short-term expansion.” How many times were the cells passaged and how long did it take? The time it takes to make a selection of the drug is critical to whether the treatment is successful. In fact an example of organotypic culture is found in the literature (Brulin et al. 2021. Evaluation of the Chemotherapy Drug Response Using Organotypic Cultures of Osteosarcoma Tumours from Mice Models and Canine Patients. Cancers 13: 4890).
Reviewer 2 Report
Comments and Suggestions for Authors
Dear Authors, your work demonstrates strong scientific merit and represents a promising direction for the future of veterinary oncology. However, I cannot recommend it for publication at this time due to the following concerns:
- The tumor-derived cells have not been sufficiently characterized, which is only acceptable for organoid models.
- The rationale behind drug selection for both groups lacks adequate foundation. Regarding the guided group, using TKI inhibitors is not informative without knowing the cells' molecular profile, and administering a non-selective drug such as doxorubicin does not contribute meaningfully to our understanding of targeted therapies.
- The study's design presents significant issues, particularly with the empirical treatment group. Here, treatment decisions were based solely on histopathological analysis, without immunohistochemical profiling. This approach undermines the validity of the outcomes, making the results in this group understandably less favorable compared to the guided group.
To make meaningful comparisons between the two groups, I recommend the following:
Drugs should be identical across groups and chosen based on a solid scientific rationale, including knowledge of the molecular profile of the target tumor (both in the literature and with experiments)
The usefulness of in vitro screening is questionable when targeted therapies like TOC are randomly assigned in the empirical group. The lack of knowledge about the molecular profile and the random selection of drugs in the empirical group prevent reliable comparisons.
Reviewer 3 Report
Comments and Suggestions for Authors
Thank you to the authors for their work in introducing the In Vitro Drug-Sensitivity platform for application in veterinary medicine. While I do not have major concerns, I do have a few inquiries:
1. The in vitro drug sensitivity testing results shown as a dose-response curve in Table I should also include a vehicle control group.
2. Lines 142-146: As various cancer cell lines from different animals were included and their proliferation rates may differ, could the authors also include information about the level of cell confluence at which the drug sensitivity testing was performed? Tight junctions in 100% confluent cells may influence the efficacy of the anticancer drugs.
3. Some data points (such as #4 in Table 1) exhibit a significant time interval between the treatment duration and TPP. Did these subjects experience adverse events or any other issues that required the treatment to be halted prematurely? Additionally, could the authors include quantification data to demonstrate the relationship between treatment duration and TTP?